# The Impact of α-Adrenoceptors in the Regulation of the Hypotonicity-Induced Increase in Duodenal Mucosal Permeability In Vivo

**DOI:** 10.3390/pharmaceutics13122096

**Published:** 2021-12-06

**Authors:** John Sedin, David Dahlgren, Markus Sjöblom, Olof Nylander

**Affiliations:** Department of Neuroscience, Division of Physiology, Uppsala University, 75236 Uppsala, Sweden; john.sedin@farmbio.uu.se (J.S.); david.dahlgren@farmbio.uu.se (D.D.); markus.sjoblom@neuro.uu.se (M.S.)

**Keywords:** luminal osmolality, mucosal permeability, α-adrenoceptor agonists, α-adrenoceptor antagonists, COX-2, clonidine, phenylephrine, yohimbine, idazoxan, prazosin

## Abstract

The duodenal mucosa is regularly exposed to a low osmolality, and recent experiments suggest that hypotonicity increases mucosal permeability in an osmolality-dependent manner. The aim was to examine whether the sympathetic nervous system, via action on α-adrenoceptors, affects the hypotonicity-induced increase in duodenal mucosal permeability. The duodenum of anaesthetised rats was perfused in vivo with a 50 mM NaCl solution in the presence of adrenergic α-adrenoceptor drugs. Studied were the effects on mucosal permeability (blood-to-lumen clearance of ^51^Cr-EDTA), arterial blood pressure, luminal alkalinisation, transepithelial fluid flux, and motility. Hypotonicity induced a six-fold increase in mucosal permeability, a response that was reversible and repeatable. The α_2_-adrenoceptor agonist clonidine abolished the hypotonicity-induced increase in mucosal permeability, reduced arterial blood pressure, inhibited duodenal motility, and decreased luminal alkalinisation. The α_2_-adrenoceptor antagonists, yohimbine and idazoxan, prevented the inhibitory effect of clonidine on the hypotonicity-induced increase in mucosal permeability. The α_1_-agonist phenylephrine or the α_1_-antagonist prazosin elicited their predicted effect on blood pressure but did not affect the hypotonicity-induced increase in mucosal permeability. None of the α_1_- or α_2_-adrenoceptor drugs changed the hypotonicity-induced net fluid absorption. In conclusion, stimulation of the adrenergic α_2_-adrenoceptor prevents the hypotonicity-induced increase in mucosal permeability, suggesting that the sympathetic nervous system has the capability to regulate duodenal mucosal permeability.

## 1. Introduction

When an individual drinks water, coffee, diet soda, or tea on an empty stomach, the duodenal mucosa will be exposed to a fluid osmolality considerably lower than in blood plasma. The low luminal osmolality induces water absorption and increases the efflux of osmolytes, which combined increase luminal osmolality. These transport processes have long been considered to occur passively across the “leaky” duodenal epithelium, by means of osmosis and solute diffusion. In 2003, Nylander and coworkers [1] demonstrated that perfusion of the duodenum with hypotonic solutions increased, in an osmolality-dependent manner, the blood-to-lumen clearance of chromium-51 labeled ethylenediamine tetraacetic acid (^51^Cr-EDTA), despite the presence of a marked net water absorption. Additional experiments revealed that luminal hypotonicity also increases the blood-to-lumen clearance of ^14^C-methylglucose and ^14^C-inulin, implicating increases in epithelial paracellular permeability [2]. Interestingly, the hypotonicity-induced increase in mucosal permeability showed to be fully reversible within 20 min after cessation of the hypotonic perfusion, supporting the view of a physiological response.

Further experiments in our laboratory showed that inhibition of cyclooxygenase (COX) activity, mainly the COX-2 isoform, markedly potentiated the hypotonicity-induced increase in mucosal permeability. This suggests that some endogenous prostanoid, presumably prostacyclin, suppresses increases in mucosal permeability [3]. Furthermore, a non-selective blockade of nicotinic acetylcholine receptors (nAChR:s) by hexamethonium abolishes the increase in mucosal permeability, implicating the involvement of a neural mechanism in the response [1]. Moreover, intravenous (iv) infusion of vasoactive intestinal peptide (VIP), a recognized neurotransmitter in the enteric nervous system, in a dose that stimulates fluid secretion and luminal alkalinisation, markedly reduced the hypotonicity-induced increase in mucosal permeability [4]. In another study, it was shown that the selective 5-hydroxytryptamine-3 (5-HT_3_) receptor antagonists attenuated, and that luminally applied 5-HT augmented the hypotonicity-induced increase in duodenal mucosal permeability, suggesting that 5-HT participates in the mediation of the response [5]. Interestingly, luminally administered lidocaine, a well-known local anaesthetic and neural blocker, at a concentration that did not affect basal ^51^Cr-EDTA clearance, strongly amplified the hypotonicity-induced increase in mucosal permeability. In fact, the 100 mOsm-induced net increase in permeability in lidocaine treated rats was the same as that induced by pure water [1]. Taken together, these results strongly suggests that the hypotonicity-induced increase in duodenal mucosal permeability is physiologically regulated. It appears that there are two main mechanisms that regulate mucosal permeability; one excitatory, which is dependent on nAChR:s, and another that is inhibitory, involving some prostanoid and possibly VIP.

The autonomic nervous system participates in the regulation of intestinal function [6]. Generally speaking, the sympathetic nervous system exerts mainly inhibitory effects on gastrointestinal motility and fluid secretion. Sympathetic nerve fibers enter the intestinal wall along arteries and terminate in the myenteric and submucosal plexuses, where they control secretomotor neurons and vascular tone via α_1_ and α_2_ adrenergic signaling. In addition, adrenergic nerve fibers also travel along blood vessels into the submucosa and mucosa in close proximity to epithelial and enterochromaffin cells, where they are believed to modulate immune responses and inflammation [7].

The aim of the present study was to investigate the effects of adrenergic α-receptors on duodenal functions, with a focus on the hypotonicity-induced increase in mucosal permeability. To accomplish this goal, the proximal duodenum of anaesthetised rats was perfused with 50 mM NaCl in the presence of α_1_- and α_2_-adrenoceptor antagonists and agonists. The effects on blood-to-lumen clearance of ^51^Cr-EDTA, arterial blood pressure, duodenal motility, transepithelial fluid flux, and luminal alkalinization were monitored.

## 2. Materials and Methods

### 2.1. Animals and Surgery

The material in this study was in conformity with Good Publishing Practice in Physiology [8]. Male Sprague Dawley rats, purchased from Taconic, Denmark, weighing 265 to 395 g, were housed in standard macrolon cages, Type IV (595 mm × 380 mm × 200 mm), in groups of two or more animals. The rats were maintained under constant conditions (temperature 20 ± 1 °C and 50 ± 10% humidity) on a 12 h light-dark cycle. All animals had free access to food and water. Before the experiments, the animals were fasted for 16 h overnight, given free access to water. To minimize the preoperative stress, all rats were anesthetized with Inactin^®^ 125 mg kg^−1^ intraperitoneally by experienced personal at the animal department, Biomedical Center, Uppsala University, Uppsala. The rat was then transported to the laboratory and immediately laid down on a pre-warmed heating pad. Body temperature was maintained at 37.5 ± 0.5 °C by a temperature regulator, connected to an intrarectal thermistor probe and a heating pad. Thereafter, the animals underwent catheterization of arteries (for assessment of mean arterial blood pressure, MABP) and veins (for infusion of ^51^Cr-EDTA and drugs), as well as abdominal surgery. The surgical procedure was the same as that described by Sedin et al. [9].

### 2.2. Measurement of Duodenal Epithelial Permeability

Determining the blood-to-lumen clearance of ^51^Cr-EDTA assessed duodenal epithelial permeability. The values are expressed as ml per min per 100 g wet tissue weight (ml min^−1^ 100 g^−1^). A detailed description of this method is found elsewhere [10,11].

### 2.3. Measurement of Duodenal Motility

Intraluminal pressure was used to assess duodenal motility. The inlet perfusion cannula was connected, via a T-tube, to a pressure transducer, and intraluminal pressure was recorded. The outlet was positioned at the same level as the inlet cannula. An upward deflection of at least 2 mmHg above baseline was defined as a motor response. The changes in intraluminal pressure were recorded, via a digitizer, on a computer using Power-Lab^®^ software (ADInstruments; Hastings, UK). Duodenal motility was assessed over intervals of 10 min, using planimetry to measure the total area under the pressure curve (AUC) during the sample period. The values given are the mean ± SEM of three 10 min intervals.

### 2.4. Measurement of Duodenal Luminal Alkalinization

The rate of luminal alkalinisation in the duodenum was determined by back titration of the perfusate to pH 5.0, with 10 mM HCl, under continuous gassing (100% N_2_), using pH-stat equipment (Schott-Titroline Easy, Mainz, Germany). The pH electrode was routinely calibrated with standard buffers before titration initiation. The amount of HCl needed to back titrate the blank solutions to pH 5.0 was negligible. The rate of luminal alkalinisation was expressed as micromoles of base transported per square centimetre serosal surface area per hour (µmol cm^−2^ h^−1^).

### 2.5. Measurement of Fluid Flux

The duodenum continuously absorbs and secretes fluid, and the difference between these two processes determines the net flux of fluid across the epithelium. The weight of the fluid in the collection vial was used to measure the flow during the 10 min collection intervals. Effluent volumes were calculated by correcting for density. The absolute net fluid flux across the mucosa was determined by subtracting the effluent volume from that delivered by the peristaltic pump alone (i.e., in the absence of the intestinal segment) and expressed as mL of fluid per gram of wet tissue weight per hour (mL g^−1^ h^−1^). The rate of the peristaltic pump was determined from the mean of two samples, obtained at 10 min intervals, immediately after terminating each experiment. A negative flux value indicates net fluid absorption. Perfusion rate was ~0.4 mL min^−1^.

### 2.6. Experimental Protocol

An illustration of the experimental protocol is presented in Figure 1. To prevent duodenal postoperative ileus, caused by the handling of the duodenum during the operative procedure, all animals were pre-treated with parecoxib, a selective COX-2 inhibitor. Previous experiments have shown that COX-2 inhibition restores motility, increases luminal alkalinization, and responds to luminal hypotonicity, with a larger increase in mucosal permeability and a greater rate of fluid absorption [12].

Parecoxib was administered iv, at a dose of 5 mg kg^−1^ 50–60 min, prior to the start of the experiment. Subsequently, ^51^Cr-EDTA was administered as an iv bolus of 75 µCi, followed by a continuously infusion, at a rate of 50 µCi h^−1^. In all animals, the duodenal segment was perfused as follows: 30 min with isotonic saline, 30 min with hypotonic saline (50 mM NaCl), 40 min recovery period with isotonic saline, 30 min with hypotonic saline, and, finally, a 30 min recovery period, with isotonic saline, giving a total perfusion time of 160 min.

The animals were divided into six groups: (1) control (n = 6), (2) clonidine (n = 6), (3) yohimbine + clonidine (n = 5), (4) idazoxan + clonidine (n = 5), (5) phenylephrine (n = 6), and (6) prazosin (n = 5). The effects on MABP, duodenal motility, luminal alkalinisation, ^51^Cr-EDTA clearance, and net fluid flux were measured.

In group 1, no drugs, besides parecoxib, were given. In group 2, and during the first 70 min of the experiment, the protocol was exactly the same as in group 1. Thereafter, clonidine was administered as an iv infusion, at a dose of 10 µg kg^−1^ h^−1^, starting 30 min prior to the second hypotonic perfusion period. In group 3, the α_2_-adrenoceptor antagonist yohimbine was administered as a slow iv bolus (1 min), at the dose 1.5 mg kg^−1^, 30 min prior to start of effluent collection. Clonidine was administered as in group 2. In group 4, idazoxan, an α_2_-adrenoceptor antagonist was administered as an iv bolus of 100 µg kg^−1^, followed by an iv infusion of 200 µg kg^−1^ h^−1^ 30 min before the start of effluent collection. Clonidine was administered as in group 2 and 3. The doses of yohimbine and idazoxan were taken from the literature [13,14,15,16]. In group 5, and during the first 70 min of the experiment, the protocol was exactly the same as in group 1. The adrenergic α_1_-adrenoceptor agonist phenylephrine was given as a slow iv infusion, at the dose of 500 µg kg^−1^ h^−1^, 30 min prior to the second hypotonic perfusion period. Finally, in group 6, the α_1_-adrenoceptor antagonist prazosin was administered as an iv bolus of 50 µg kg^−1^, followed by an iv infusion of 50 µg kg^−1^ h^−1^, 30 min prior to the second hypotonic perfusion period. Doses of drugs were chosen from previous studies [17,18].

### 2.7. Chemicals

Bovine albumin, yohimbine HCl, phenylephrine HCl, and idazoxan HCl were obtained from Sigma Chemicals (St. Louis, MO, USA). Clonidine HCl and prazosin HCl were purchased from Tocris Bioscience (Bristol, UK). ^51^Cr-EDTA was purchased from PerkinElmer Life Sciences Inc., Boston, MA, USA. Thiobutabarbital sodium salt (Inactin^®^) was obtained from RBI, Natick, MA, USA. Parecoxib (Dynastat® for injection) was purchased from Pfizer Inc., New York, NY, USA.

### 2.8. Statistics

Descriptive statistics are expressed as mean ± SEM. The statistical significance of data was tested by analysis of variance (ANOVA). Results obtained before, during, and after perfusion with hypotonic solution were compared by one-way repeated-measures ANOVA, followed by Tukey’s multiple comparison test (within group), and by two-way repeated-measure ANOVA (mixed-model), followed by Bonferroni post-tests (between groups). All statistical analyses were performed on an Apple^®^ computer, using GraphPad Prism software. A P-value less than 0.05 was considered significant.

## 3. Results

### 3.1. Basal Parameters

The median and min to max of basal values, obtained during the perfusion with isotonic saline, for group 1, 2, 5, and 6, are presented (as box plots with whiskers) in Figure 2a–e (n = 22–24). In Figure 2f, the net increase in ^51^Cr-EDTA clearance, in response to luminal hypotonicity, is shown.

### 3.2. Control Rats, MABP, ^51^Cr-EDTA Clearance, Luminal Alkalinisation, Net Fluid Absorption, and Duodenal Motility

MABP was 106 ± 3 mmHg at the beginning of the experiment and decreased to 82 ± 7 mmHg (P < 0.05) 160 min later. During the initial 30 min perfusion with isotonic saline, the basal ^51^Cr-EDTA clearance, luminal alkalinisation, and duodenal motility were 0.35 ± 0.02 mL min^−1^ 100 g^−1^, 5.5 ± 0.4 µmol cm^−2^ h^−1^, and 700 ± 119 AUC 10 min^−1^, respectively. The basal mean net fluid flux was 0.75 ± 0.15 mL g^−1^ h^−1^ (Figure 3a). Perfusion of the duodenum with 50 mM NaCl induced net fluid absorption and increased ^51^Cr-EDTA clearance 5.5-fold (P < 0.001), which returned to basal values within 20 min after cessation of the hypotonic saline perfusion (Figure 3a and Figure 4a). The net increase in ^51^Cr-EDTA clearance, in response to luminal hypotonicity, was 1.58 ± 0.23 mL min^−1^ 100 g^−1^. The second perfusion with 50 mM NaCl, starting 40 min after cessation of the first one, induced an increase in ^51^Cr-EDTA (1.88 ± 0.25 mL min^−1^ 100 g^−1^) of similar magnitude as the first one (Figure 4a). The change in net fluid flux, in response to the second hypotonic perfusion, was almost identical to the first one (Figure 3a). Luminal hypotonicity had no effect on either luminal alkalinisation or duodenal motility (Figure 4b,c).

### 3.3. Effects of Clonidine (α_2_-Adrenoceptor Agonist)

During the initial 30 min perfusion with isotonic saline, the MABP was 94 ± 4 mmHg, which declined to 83 ± 5 mmHg just prior to the infusion of clonidine. Basal ^51^Cr-EDTA clearance, luminal alkalinisation, and duodenal motility were 0.26 ± 0.02 mL min^−1^ 100 g^−1^, 7.3 ± 0.7 µmol cm^−2^ h^−1^, and 730 ± 158 AUC 10 min^−1^, respectively. The mean net fluid flux was −0.12 ± 0.12 mL g^−1^ h^−1^. The net increase in ^51^Cr-EDTA clearance, in response to perfusion with 50 mM NaCl, was 1.88 ± 0.25 mL min^−1^ 100 g^−1^, a value not different from that obtained in group 1 (Figure 4a). Clonidine infusion reduced MABP to 61 ± 3 mm Hg (P < 0.05). During the perfusion with isotonic saline, in the period between the hypotonic perfusions, clonidine induced (P < 0.05) net fluid absorption (Figure 3a). Furthermore, clonidine abolished duodenal motility (P < 0.01) and significantly (P < 0.001) decreased luminal luminal alkalinisation (Figure 4b,c). Treatment with clonidine abolished (P < 0.01) the 50 mM NaCl-induced increase in ^51^Cr-EDTA clearance (Figure 4a), but the change in net fluid flux was not different from that in the control group (Figure 3a).

### 3.4. Effects of Yohimbine + Clonidine (α_2_-Adrenoceptor Antagonist + α_2_-Adrenoceptor Agonist)

In yohimbine pre-treated animals, the MABP was 96 ± 5 mmHg in the beginning of the experiment and 103 ± 4 mmHg 70 min later, i.e., prior to the infusion of clonidine. Basal ^51^Cr-EDTA clearance (0.26 ± 0.05 mL min^−1^ g^−1^), luminal alkalinisation (8.3 ± 0.8 µmol cm^−2^ h^−1^), duodenal motility (483 ± 151 AUC 10 min^−1^), and net fluid flux (−0.87 ± 0.24 mL g^−1^ h^−1^) were not different from the corresponding values in the controls (Figure 3 and Figure 5). Luminal perfusion of the duodenum with 50 mM NaCl increased ^51^Cr-EDTA clearance and the net increase was 2.35 ± 0.23 mL min^−1^ 100 g^−1^, significantly (P < 0.05) higher than the corresponding value in the control group (Figure 5a). Clonidine tended to reduce MABP in yohimbine-treated animals, but this effect did not attain statistical significance (data not shown). The net increase in ^51^Cr-EDTA clearance, in response to luminal hypotonicity in yohimbine, plus clonidine-treated animals (1.45 ± 0.30 mL min^−1^ 100 g^−1^), was not different from controls but significantly higher (P < 0.05) than in animals treated with clonidine alone (Figure 5a). Clonidine infusion attenuated (P < 0.05), but did not abolish duodenal motility, and reduced the luminal alkalinisation in yohimbine-treated animals (Figure 5b,c). Clonidine did not affect the hypotonicity-induced net fluid flux in animals treated with yohimbine (Figure 3b).

### 3.5. Effect of Idazoxan + Clonidine (α_2_-Adrenoceptor Antagonist +Aagonist)

The MABP was 108 ± 9 mmHg at the beginning of the experiment and 91 ± 10 mmHg immediately before the administration of clonidine. Basal ^51^Cr-EDTA clearance (0.38 ± 0.05 mL min^−1^ 100 g^−1^), luminal alkalinisation (7.1 ± 1.0 µmol cm^−2^ h^−1^), duodenal motility, and net fluid flux (−0.24 ± 0.21 mL g^−1^ h^−1^) were not different from the corresponding values in the control group. Luminal perfusion of the duodenum with 50 mM NaCl increased ^51^Cr-EDTA clearance (Figure 5a). The net increase in ^51^Cr-EDTA clearance was 1.62 ± 0.23 mL min^−1^ 100 g^−1^, a value not different from the corresponding net increase in the control group. Clonidine did not affect MABP in idazoxan-treated animals (data not shown). The net increase in ^51^Cr-EDTA clearance (1.98 ± 0.35 mL min^−1^ 100 g^−1^), in response to luminal hypotonicity in idazoxan + clonidine-treated animals, was not different from the corresponding value in the control group. In idazoxan-treated animals clonidine did not affect luminal alkalinisation and, furthermore, prevented the clonidine-induced inhibition of duodenal motility (P < 0.01) (Figure 5b,c).

### 3.6. Effects of Phenylephrine (α_1_-Adrenoceptor Agonist)

The MABP was 98 ± 3 mmHg at the beginning of the experiment and 92 ± 2 mmHg immediately before the start of the iv infusion of phenylephrine. Basal values, during the initial 30 min period of the isotonic saline perfusion, were 0.28 ± 0.05 mL min^−1^ 100 g^−1^, 6.5 ± 1.5 µmol cm^−2^ h^−1^, and 472 ± 85 AUC per 10 min and 0.19 ± 0.48 mL g^−1^ h^−1^ for ^51^Cr-EDTA clearance, luminal alkalinisation, motility, and net fluid flux, respectively. The net increase in ^51^Cr-EDTA clearance, in response to luminal perfusion with 50 mM NaCl, was 1.57 ± 0.37 mL min^−1^ 100 g^−1^. (Figure 6a). Intravenous infusion of phenylephrine significantly increased MABP from 92 ± 2 to 116 ± 5 mmHg (P < 0.001). In the presence of phenylephrine, luminal perfusion with 50 mM NaCl markedly increased ^51^Cr-EDTA clearance (1.72 ± 0.51 mL min^−1^ 100 g^−1^), a value significantly larger than that in clonidine-treated animals but not different from that in controls (Figure 6a). Phenylephrine did not affect luminal alkalinisation or duodenal motility (Figure 6b,c).

### 3.7. Effects of Prazosin (α_1_-Adrenoceptor Antagonist)

The MABP was 105 ± 5 mmHg at the beginning of the experiment and 98 ± 2 mmHg immediately before the iv injection of prazosin. Basal values, during the initial 30 min period of the isotonic saline perfusion, were 0.29 ± 0.06 mL min^−1^ 100 g^−1^, 6.2 ± 1.0 µmol cm^−2^ h^−1^, 517 ± 177 AUC per 10 min and −0.49 ± 0.53 mL g^−1^ h^−1^ for ^51^Cr-EDTA clearance, luminal alkalinisation, motility, and net fluid flux, respectively. The net increase in ^51^Cr-EDTA clearance, in response to luminal perfusion with 50 mM NaCl, was 1.31 ± 0.18 mL min^−1^ 100 g^−1^, a value not different from controls (Figure 6a). Injection of prazosin decreased MABP from 98 ± 2 to 68 ± 3 mmHg (P < 0.001). In the presence of prazosin, luminal perfusion with 50 mM NaCl markedly increased ^51^Cr-EDTA clearance, and the net increase was 1.73 ± 0.32 mL min^−1^ 100 g^−1^, a value not different from controls (Figure 6a). Prazosin did not affect luminal alkalinisation, duodenal motility, or net fluid flux (Figure 6b,c and Figure 3c, respectively).

## 4. Discussion

Previous experiments from our laboratory demonstrate that luminal hypotonic solutions increase duodenal mucosal permeability in an osmolality-dependent manner, i.e., the lower the luminal osmolality, the greater the increase in mucosal permeability. Furthermore, recent data strongly suggest that the increase in mucosal permeability is physiological regulated. The major aim of the present investigation was to elucidate whether α-adrenergic receptor agonists and antagonists affect the hypotonicity-induced increase in duodenal mucosal permeability. For that purpose, we used the α_2_-adrenoceptor agonist clonidine, α_2_-adrenoceptor antagonists yohimbine and idazoxan, α_1_-adrenoceptor agonist phenylephrine, and α_1_-adrenoceptor antagonist prazosin. The osmolality of the perfusion solution (100 mOsmol kg^−1^ H_2_O) is clearly within the physiological range, at least in humans [19]. For comparison, the effects of α-adrenergic receptor agonists and antagonists on duodenal luminal alkalinisation, motility, transepithelial net fluid flux, and MABP were also assessed.

In this investigation, duodenal mucosal permeability was determined by measuring the flux of ^51^Cr-EDTA from blood-to-lumen. ^51^Cr-EDTA was administered iv at a constant rate throughout the experiment, and the ^51^Cr-activity assessed in the duodenal lumen and blood plasma. ^51^Cr-EDTA has to cross the capillary wall, interstitium, and epithelium before reaching the duodenal lumen. Previous data strongly suggests that the rate-limiting barrier for the blood-to-lumen movement of ^51^Cr-EDTA is the intestinal epithelium [20]. Furthermore, the hydrophilic properties [21] and cross-sectional radius of Cr-EDTA [22] suggests that the paracellular pathway constitutes the predominant route of transepithelial passage for this tracer.

Here, we confirm previous findings that luminal perfusion of the duodenum, with a 50 mM NaCl solution for 30 min, markedly increased the blood-to-lumen clearance of ^51^Cr-EDTA, implying increases in duodenal mucosal paracellular permeability. A second luminal perfusion with an identical hypotonic solution, starting 40 min after cessation of the first one, induced virtually, the same increase in mucosal permeability as the first one, demonstrating excellent reproducibility. Moreover, in both instances, the recovery of mucosal permeability after the cessation of the hypotonic perfusion was fast, within 20 min, significative of a physiological response.

Intravenous infusion of clonidine abolished the hypotonicity-induced increase in duodenal mucosal permeability. This effect of clonidine was prevented by α_2_-adrenoceptor antagonist, yohimbine, and idazoxan, suggesting that clonidine abolishes the hypotonicity-induced increase in mucosal permeability via stimulation of α_2_-adrenoceptors. Interestingly, iv administration of the α_1_-adrenoceptor agonist phenylephrine or the α_1_-adrenoceptor antagonist prazosin exerted their predicted effect on arterial blood pressure but had virtually no effect on any of the duodenal functions studied. Hence, these data raise the possibility that increased activity in sympathetic nerves to the duodenum, via an action of noradrenalin on α_2_-adrenoceptors, have the capability of reducing the hypotonicity-induced increase in mucosal permeability.

So, what is the mechanism by which clonidine inhibits the hypotonicity-induced increase in mucosal permeability? In a study by Ahsan et al. [23], in isolated rabbit ileum, it was suggested that α_2_-agonists reduce interstitial fluid pressure by depressing smooth muscle tone. A decrease in interstitial fluid pressure may reduce the driving forces for the transport of ^51^Cr-EDTA from interstitium-to-lumen. Previous data, and those in the present study, indicate that the magnitude of the hypotonicity-induced increase in mucosal permeability may be coupled to duodenal smooth muscle activity. Indeed, experiments in rat duodenum have shown that COX-inhibition, which induces duodenal motility, results in a greater increase in the hypotonicity-induced increase in mucosal permeability, compared to control animals lacking contraction [1]. Administration of iloprost, a prostacyclin analogue, reversed the response of COX inhibition on both motility and the hypotonicity-induced increase in mucosal permeability, suggesting that prostacyclin is responsible. Moreover, in the same study, it was shown that blockade of nicotinic receptors by hexamethonium markedly reduced duodenal motility and, at the same time, abolished the hypotonicity-induced increase in mucosal permeability. In the present study, it was shown that the α-adrenoceptors antagonists idazoxan and prazosin, or the agonist phenylephrine, neither effected the parecoxib-induced duodenal motility nor the hypotonicity-induced increase in mucosal permeability. In contrast, clonidine abolished duodenal motility and markedly reduced the hypotonicity-induced increase in mucosal permeability. Although these results support a close coupling between increased duodenal smooth muscle activity and hypotonicity-induced increase in mucosal permeability, it should be noted that increased duodenal motility does not increase basal mucosal permeability per se [24]. Moreover, it has previously been shown that iv infusion of VIP, a neuropeptide released from enteric neurons, markedly reduces, similar to clonidine, the hypotonicity-induced increase in mucosal permeability but without affecting duodenal motility [4]. Furthermore, it has been shown that the nicotinic receptor antagonist hexamethonium abolished duodenal motility but did not affect the ethanol-induced increase in duodenal mucosal permeability [25]. It, therefore, seems unlikely that attenuation of duodenal smooth muscle tone by clonidine is the major causative factor inhibiting the hypotonicity-induced increase in mucosal permeability.

The inhibitory effect of clonidine on duodenal motility is in agreement with previous findings in conscious rats [24]. It is generally believed that clonidine reduces arterial blood pressure by a central mechanism, at least in part, via inhibition of sympathetic tone to the vasculature, thereby reducing peripheral resistance [14,26]. The inhibitory effect of clonidine on duodenal motility occurred within minutes after onset of the iv infusion, well before any effect on MABP was seen, suggesting a peripheral effect of clonidine, as well. Since small intestinal motility is regulated by neural mechanisms, it seems likely that clonidine exerts its inhibitory action on duodenal motility via depression of excitatory motor neurons in the enteric nervous system. The prevailing view is that this occurs via activation of inhibitory α_2_-adrenoceptors on cholinergic nerve terminals in muscularis externa [6].

Another possibility is that clonidine attenuates the hypotonicity-induced increase in mucosal permeability by reducing duodenal mucosal blood flow via an α-adrenoceptor-induced arteriole vasoconstriction. Arteriole vasoconstriction may impede fluid filtration across the capillary wall, thereby reducing interstitial fluid pressure, which, in turn, reduces paracellular fluid transport. However, here we show that the α_1_-adrenoceptor agonist phenylephrine, a potent vasoconstrictor, did not affect the hypotonicity-induced increase in mucosal permeability. Furthermore, previous experiments in rat duodenum have shown that phenylephrine increases vascular resistance and reduces blood flow by about 20% [25]. Interestingly, the α_1_-adrenoceptor antagonist prazosin decreased the mean arterial blood pressure, as did clonidine, without affecting the hypotonicity-induced increase in duodenal mucosal permeability. Additionally, prazosin have been shown to decrease both vascular resistance and blood flow in rat duodenum [25]. Hence, it seems highly unlikely that the hypotonicity-induced increase in mucosal permeability is related to changes in vascular resistance or blood flow.

Based on these data, we, therefore, need to look for another mechanism of action of clonidine in inhibiting the hypotonicity-induced increase in mucosal permeability. As expected, luminal hypotonicity markedly increases the absorption of water. It seems reasonable to assume that water absorption causes the enterocytes in the villous enterocytes to swell, which may be sensed by intrinsic primary neurons beneath the epithelium. This, in turn, could elicit an intramural reflex, involving the activation of a population of enteric neurons innervating the crypts, resulting in increased paracellular solute permeability (Figure 7).

The view of activation of an intramural reflex is indirectly supported by the finding in rat duodenum that 50 mM NaCl increases the release of 5-HT, presumably from enterochromaffin cells, and that 5-HT_3_-receptor antagonists attenuate the hypotonicity-induced increase in mucosal permeability. Furthermore, the ganglionic blocker hexamethonium, as well as VIP, abolishes the hypotonicity-induced increase in duodenal mucosal permeability. The physiological function of dilation of the paracellular pathways may be to increase the interstitium-to-lumen diffusion of solutes, such as sodium, which thereby facilitates the adjustment of luminal osmolality [1,4]. Clonidine, acting on α_2_-adrenoceptors on cholinergic nerve terminals or, less likely, via direct effect on epithelial cells [27,28,29], may inhibit the intracellular signal transmission that mediates the increase in tight junction permeability. Activation of epithelial myosin-light chain kinase has been proposed as a critical regulator of increased tight junction permeability [30], and it would, therefore, be of great interest to test whether their activity is affected by luminal hypotonicity and clonidine.

## 5. Conclusions

Luminal hypotonicity markedly and reversibly increases duodenal mucosal permeability, an effect possibly aimed to increase the blood-to-lumen transport of Na^+^, which contributes to an increased luminal osmolality. We demonstrated that stimulation of adrenergic α_2_-, but not α_1_-receptors, abolishes the hypotonicity-induced increase in mucosal permeability. As the sympathetic nervous system modulates a variety of gastrointestinal functions, it seems likely that it also participates in the regulation of duodenal mucosal permeability. The physiological function of this effect is not known, but it is possible that in life-threatening situations or in response to acute stress, when the body needs to maintain an adequate extracellular fluid and blood volume, the central nervous system, via activation of the sympathetic nervous system, down-regulates some functions in the gastrointestinal tract. The purpose may be to reduce energy expenditures in the gut, for instance, by inhibiting motility and electrolyte fluid secretion, as well as tightening of the epithelial barrier.

## Figures and Tables

**Figure 1 pharmaceutics-13-02096-f001:**
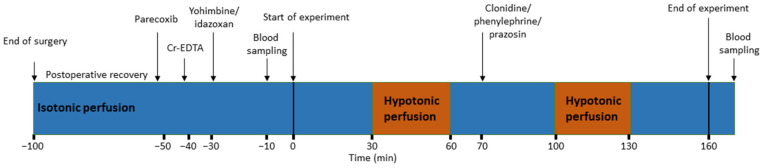
Schematic illustration of the drug administrations, events, and perfusion solutions (blue or red). The reader is referred to the method section for details.

**Figure 2 pharmaceutics-13-02096-f002:**
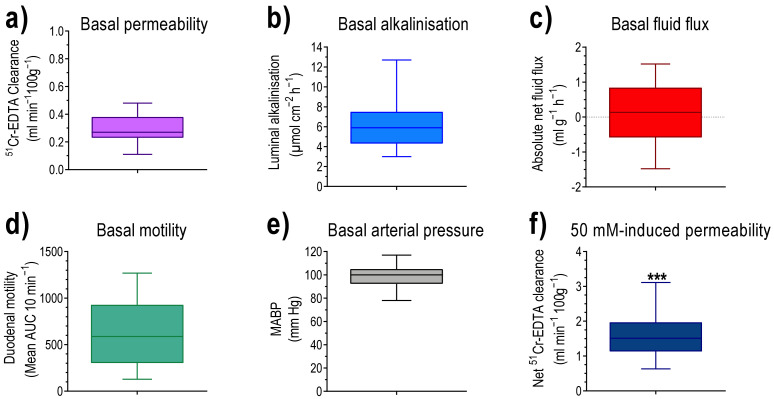
Basal values, in response to luminal perfusion with isotonic saline during the initial 30 min of the experiment (n = 22–24). (**a**) Blood-to-lumen clearance of ^51^Cr-EDTA. (**b**) Duodenal luminal alkalinisation. (**c**) Transepithelial net fluid flux. (**d**) Duodenal motility. (**e**) Mean arterial blood pressure (MABP). (**f**) The net increase (the mean of the 50 and 60 min values, minus the mean of the three basal values during the perfusion with isotonic saline) in ^51^Cr-EDTA clearance, in response to luminal perfusion of the duodenum with 50 mM NaCl. *** P < 0.001 compared with basal permeability (**a**).

**Figure 3 pharmaceutics-13-02096-f003:**
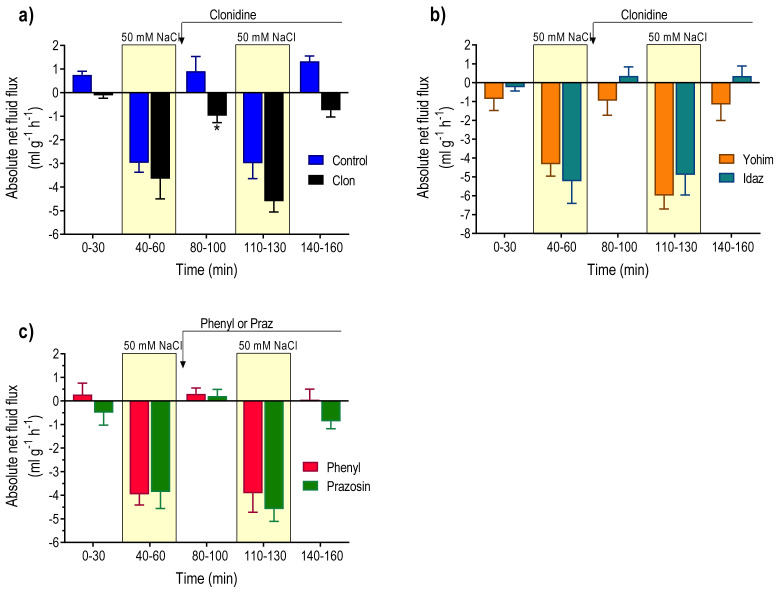
Effects of α-adrenoreceptor agonists and antagonist on the absolute net fluid flux in the rat duodenum, where a minus sign indicates fluid absorption. The duodenum was perfused with isotonic saline (white area) and 50 mM NaCl hypotonic saline (yellow bar). (**a**) Controls (n = 6) and effects of clonidine (n = 6). (**b**) Effects of an idazoxan (n = 5) or yohimbine (n = 5) in the absence and presence of clonidine. (**c**) Effects of phenylephrine (n = 6) or prazosin (n = 5). Values are mean ± SEM. * P < 0.05, compared with the mean net fluid flux at time point 0–30 min (black bar). The reader is referred to the method section for details of the experimental setup and doses.

**Figure 4 pharmaceutics-13-02096-f004:**
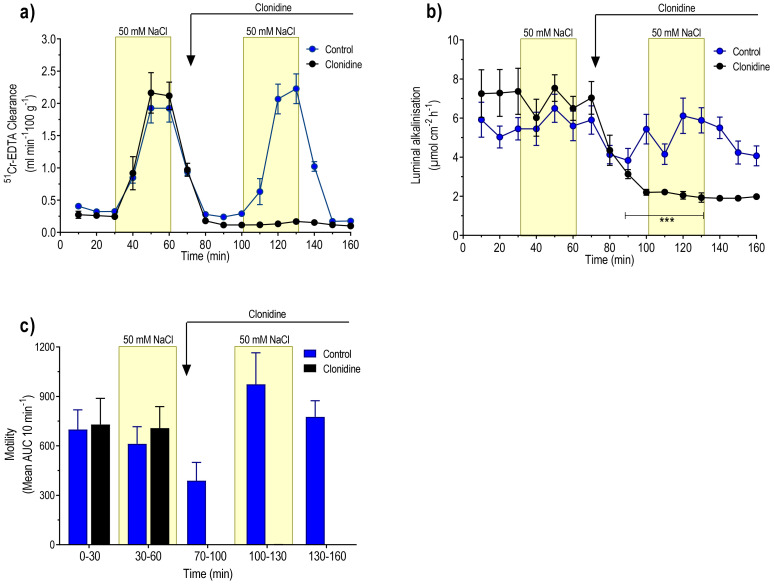
Effects of absence or presence of clonidine on (**a**) blood-to-lumen clearance of ^51^Cr-EDTA, (**b**) luminal alkalinization, and (**c**) duodenal motility. The duodenum was perfused with isotonic saline (white area) and 50 mM NaCl hypotonic saline (yellow bar). Values are mean ± SEM, n = 6 in each group. *** P < 0.001, compared with the 10–70 min values in the clonidine group (one-way, repeated-measure ANOVA, followed by Tukey’s multiple comparison test). The reader is referred to the method section for details of the experimental setup and doses.

**Figure 5 pharmaceutics-13-02096-f005:**
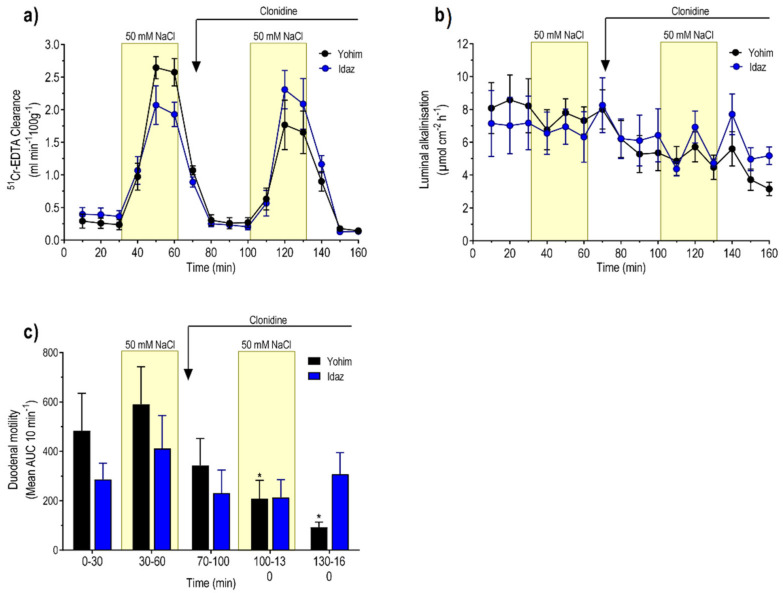
Effects of α2-adrenoreceptor antagonists (yohimbine and idazoxan), with or without clonidine, on (**a**) blood-to-lumen clearance of ^51^Cr-EDTA, (**b**) luminal alkalinisation, and (**c**) duodenal motility. The duodenum was perfused with isotonic saline (white area) and 50 mM NaCl hypotonic saline (yellow bar). Values are mean ± SEM, n = 5 in each group. * P < 0.05, compared with the values at time points 0–30 and 30–60 min (black bar). The reader is referred to the method section for details of the experimental setup and doses.

**Figure 6 pharmaceutics-13-02096-f006:**
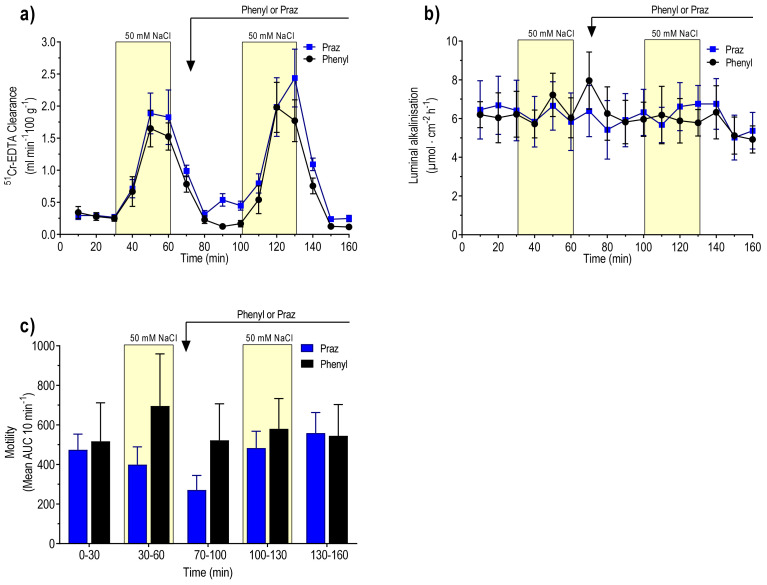
Effects of α_1_-adrenoceptor agonist (phenylephrine, n = 6) and antagonist (prazosin, n = 5) on (**a**) blood-to-lumen clearance of ^51^Cr-EDTA, (**b**) luminal alkalinization, and (**c**) duodenal motility. The duodenum was perfused with isotonic saline (white area) and 50 mM NaCl hypotonic saline (yellow bar). Values are mean ± SEM. P < 0.05, compared with the values at time points 0–30 and 30–60 min (black bar). The reader is referred to the method section for details of the experimental setup and doses.

**Figure 7 pharmaceutics-13-02096-f007:**
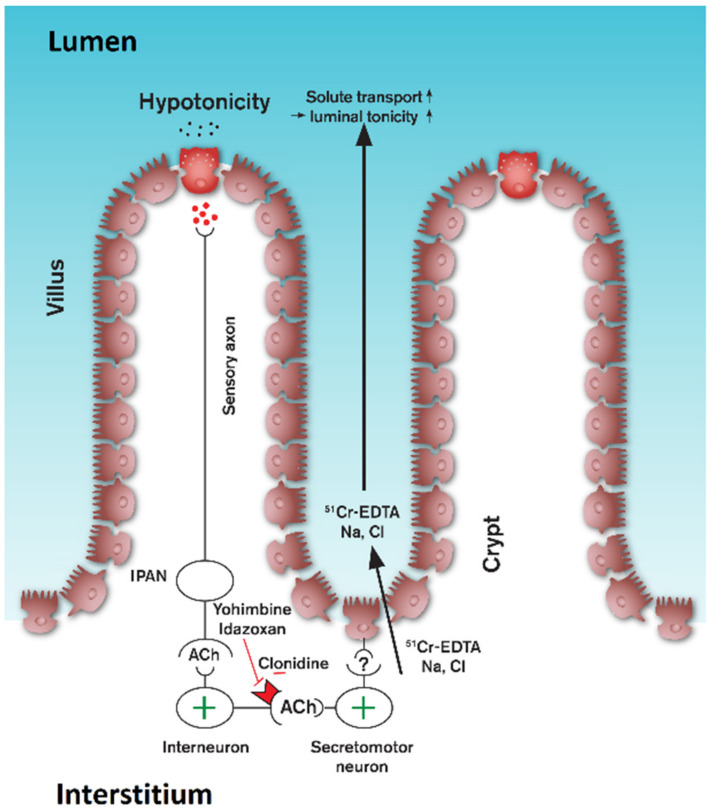
Our proposal for the mechanism by which clonidine inhibits the hypotonicity-induced increase in mucosal permeability via stimulation of the α_2_-adrenoceptor (autoreceptor). This, in turn, leads to inhibition of the hypotonicity-induced stimulation of secretomotor neurons.

## Data Availability

The data presented in this study are available on request from the corresponding author.

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
