# Peer review of "The Impact of α-Adrenoceptors in the Regulation of the Hypotonicity-Induced Increase in Duodenal Mucosal Permeability In Vivo"

_pharmaceutics, 2021, doi:10.3390/pharmaceutics13122096_

Round 1
Reviewer 1 Report
The authors have conducted an excellent set of experiments targeted to determine the role of alpha1 and alpha2 adrenoceptors on the hypotonicity-induced increase in duodenal mucosal permeability.
The doses of antagonists and specific agonists are appropriate and defended in the methods.
The authors conclude that stimulation of the alpha2-adrenoceptor prevents the hypotonicity-induced increase in mucosal permeablity via presynaptic activation of inhibitory secretomotor neurones.
The only missing tale piece is whether this mechanism is physiologically activated. The authors adequately address this issue in discussion.
Reviewer 2 Report
The current manuscript examined the effect of adrenergic α1- and α2-adrenoceptors on the hypotonicity-induced increase in duodenal mucosal permeability. 51Cr-EDTA was chosen to investigate the blood-to-lumen clearance. In addition, arterial blood pressure, luminal alkalinisation, transepithelial fluid flux and duodenal motility were determined. The current findings provided evidence that hypotonicity-induced increase in duodenal mucosal permeability could be abolished by clonidine via α2-adrenoceptor stimulation. However, duodenal mucosal permeability did not play a critical role in drug absorption. How was the permeability changed for the more important intestinal sections such as jejunum and ileum? The researchers in biopharmaceutics field will be more interested in this topic. Moreover, the authors focused on blood-to-lumen clearance, how was the lumen-to-blood absorption changed? In addition, several typical pharmaceutical substrates including low, medium and high permeability properties should be included other than 51Cr-EDTA for Pharmaceutics. As such, the current manuscript was not suitable for this journal.
Reviewer 3 Report
General comments:
- The authors did a great effort to study the effect of clonidine in diminishing the hypotonicity-induced increase in mucosal permeability via α2-adrenoceptor stimulation.
- Generally, the manuscript needs rearrangement of many sectors and position of figures to make the manuscript information and data easier in its flowability and illustrate the insights of the authors.
- General language editing is required throughout the manuscript.
- The title did not describe the experimental work in the manuscript and needs rephrasing to be matched with the article content. Not written in declarative sentences.
- Throughout the manuscript any abbreviation needs to be mentioned in full terms at its first appearance within the manuscript….example 51Cr-EDTA clearance…Chromium 51 labeled ethylene diamine tetraacetic acid, Ci ????
- All texts and paragraphs under figures captions should be inserted within the main manuscript in its right position….for example Figure 1 lines 125-131…part of it needs to be added to the experimental section and so on…the authors can illustrate their work in a more easy manner for readers.
Detailed Comments:
Abstract:
Needs general improvement and clarifying the main purpose of the article and the of the hypotonicity on general health.
Line 22 .. None of the α1- or α2-adrenoceptor drugs affected the hypotonicity-induced net fluid absorption…this is not matched with the stated results on page 5 figure 3 L 183-195.
Materials and Methods:
L 101…please mention the method in detail..ref.9 does not mention it as the authors said.
L 104 …ref.12…the same as ref 9.
L 115 … LA ??? refer to my general comments (no 5)
L 133 how many animals in each group.
L 170…perfusion with hypertonic solution??? I think the authors used hypotonic one…and we can stop here and ask…what we infuse…hyper or hypotonic one..and subsequently discuss lumen perfusion.
Results:
3.2. Controls…..change this subtitle to one which matches its result content…MABP Cr-EDTA, LA, Net fluid absorption, and duodenal motility.
Figure 3 position is following 3.2. controls….its text should be added to the main manuscript
Please clarify, what is the difference between absolute net fluid flux and net fluid absorption??
L 236…yohimbine ± clonidine ??? why the authors use ± sign ??? yohimbine pretreatment was followed by clonidine administration…no animals continued the study without clonidine administration. The same for idazoxan L 264 (refer to figure 5)
Discussion:
The authors focused on clonidine and α2 agonist with no discussion of α1 agonists or antagonists in both.
What is the rationale for using both yohimbine and idazoxan as α2 antagonists?
Figure 7 can be improved and more illustrative…Hypotonicity with an arrow?? internal and external segments???
Conclusion:
α1 agonist and antagonist effects…please mention that

Round 2
Reviewer 3 Report
Many thanks for authors to respond to our comments and the manuscript is really improved, but still needs minor revisions General minor language editing....here some examples: Line 17...reversible...not reversable. L.55...In nother study, .... L.65 .. regulate not regulates... L. 70 and 73....Fibers....not fibres L. 77...with a focus on L.354... that increased...remove an...358...isolated...remove an Line 121 Bullets and numbering was missed... 2.5. Measurement of fluid flux ???...and consider subsequent numbering of subtitles. - I think addition of subtitles to discussion part may be more valuable and increase readability of the manuscript